# Adaptive Low-Complexity Sequential Inference for Dirichlet Process Mixture Models

**Theodoros Tsiligkaridis,  Keith W. Forsythe**
Massachusetts Institute of Technology, Lincoln Laboratory
Lexington, MA 02421 USA
ttsili@ll.mit.edu, forsythe@ll.mit.edu

## Abstract

We develop a sequential low-complexity inference procedure for Dirichlet process mixtures of Gaussians for online clustering and parameter estimation when the number of clusters are unknown a-priori. We present an easily computable, closed form parametric expression for the conditional likelihood, in which hyper-parameters are recursively updated as a function of the streaming data assuming conjugate priors. Motivated by large-sample asymptotics, we propose a novel adaptive low-complexity design for the Dirichlet process concentration parameter and show that the number of classes grow at most at a logarithmic rate. We further prove that in the large-sample limit, the conditional likelihood and data predictive distribution become asymptotically Gaussian. We demonstrate through experiments on synthetic and real data sets that our approach is superior to other online state-of-the-art methods.

## 1  Introduction

Dirichlet process mixture models (DPMM) have been widely used for clustering data Neal (1992); Rasmussen (2000). Traditional finite mixture models often suffer from overfitting or underfitting of data due to possible mismatch between the model complexity and amount of data. Thus, model selection or model averaging is required to find the correct number of clusters or the model with the appropriate complexity. This requires significant computation for high-dimensional data sets or large samples. Bayesian nonparametric modeling are alternative approaches to parametric modeling, an example being DPMM's which can automatically infer the number of clusters from the data via Bayesian inference techniques.

The use of Markov chain Monte Carlo (MCMC) methods for Dirichlet process mixtures has made inference tractable Neal (2000). However, these methods can exhibit slow convergence and their convergence can be tough to detect. Alternatives include variational methods Blei & Jordan (2006), which are deterministic algorithms that convert inference to optimization. These approaches can take a significant computational effort even for moderate sized data sets. For large-scale data sets and low-latency applications with streaming data, there is a need for inference algorithms that are much faster and do not require multiple passes through the data. In this work, we focus on low-complexity algorithms that adapt to each sample as they arrive, making them highly scalable. An online algorithm for learning DPMM's based on a sequential variational approximation (SVA) was proposed in Lin (2013), and the authors in Wang & Dunson (2011) recently proposed a sequential maximum *a-posterior* (MAP) estimator for the class labels given streaming data. The algorithm is called sequential updating and greedy search (SUGS) and each iteration is composed of a greedy selection step and a posterior update step.

The choice of concentration parameter $\alpha$ is critical for DPMM's as it controls the number of clusters Antoniak (1974). While most fast DPMM algorithms use a fixed $\alpha$ Fearnhead (2004); Daume

(2007); Kurihara et al. (2006), imposing a prior distribution on $\alpha$ and sampling from it provides more flexibility, but this approach still heavily relies on experimentation and prior knowledge. Thus, many fast inference methods for Dirichlet process mixture models have been proposed that can adapt $\alpha$ to the data, including the works Escobar & West (1995) where learning of $\alpha$ is incorporated in the Gibbs sampling analysis, Blei & Jordan (2006) where a Gamma prior is used in a conjugate manner directly in the variational inference algorithm. Wang & Dunson (2011) also account for model uncertainty on the concentration parameter $\alpha$ in a Bayesian manner directly in the sequential inference procedure. This approach can be computationally expensive, as discretization of the domain of $\alpha$ is needed, and its stability highly depends on the initial distribution on $\alpha$ and on the range of values of $\alpha$. To the best of our knowledge, we are the first to analytically study the evolution and stability of the adapted sequence of $\alpha$'s in the online learning setting.

In this paper, we propose an adaptive non-Bayesian approach for adapting $\alpha$ motivated by large-sample asymptotics, and call the resulting algorithm ASUGS (Adaptive SUGS). While the basic idea behind ASUGS is directly related to the greedy approach of SUGS, the main contribution is a novel low-complexity stable method for choosing the concentration parameter adaptively as new data arrive, which greatly improves the clustering performance. We derive an upper bound on the number of classes, logarithmic in the number of samples, and further prove that the sequence of concentration parameters that results from this adaptive design is almost bounded. We finally prove, that the conditional likelihood, which is the primary tool used for Bayesian-based online clustering, is asymptotically Gaussian in the large-sample limit, implying that the clustering part of ASUGS asymptotically behaves as a Gaussian classifier. Experiments show that our method outperforms other state-of-the-art methods for online learning of DPMM's.

The paper is organized as follows. In Section 2, we review the sequential inference framework for DPMM's that we will build upon, introduce notation and propose our adaptive modification. In Section 3, the probabilistic data model is given and sequential inference steps are shown. Section 4 contains the growth rate analysis of the number of classes and the adaptively-designed concentration parameters, and Section 5 contains the Gaussian large-sample approximation to the conditional likelihood. Experimental results are shown in Section 6 and we conclude in Section 7.

## 2   Sequential Inference Framework for DPMM

Here, we review the SUGS framework of Wang & Dunson (2011) for online clustering. Here, the nonparametric nature of the Dirichlet process manifests itself as modeling mixture models with countably infinite components. Let the observations be given by $\mathbf{y}_i \in \mathbb{R}^d$, and $\gamma_i$ to denote the class label of the $i$th observation (a latent variable). We define the available information at time $i$ as $\mathbf{y}^{(i)} = \{\mathbf{y}_1, \ldots, \mathbf{y}_i\}$ and $\gamma^{(i-1)} = \{\gamma_1, \ldots, \gamma_{i-1}\}$. The online sequential updating and greedy search (SUGS) algorithm is summarized next for completeness. Set $\gamma_1 = 1$ and calculate $\pi(\theta_1|\mathbf{y}_1, \gamma_1)$. For $i \geq 2$,

1. Choose best class label for $\mathbf{y}_i$:   $\gamma_i \in \arg\max_{1 \leq h \leq k_{i-1}+1} P(\gamma_i = h|\mathbf{y}^{(i)}, \gamma^{(i-1)})$.

2. Update the posterior distribution using $\mathbf{y}_i, \gamma_i$:   $\pi(\theta_{\gamma_i}|\mathbf{y}^{(i)}, \gamma^{(i)}) \propto f(\mathbf{y}_i|\theta_{\gamma_i})\pi(\theta_{\gamma_i}|\mathbf{y}^{(i-1)}, \gamma^{(i-1)})$.

where $\theta_h$ are the parameters of class $h$, $f(\mathbf{y}_i|\theta_h)$ is the observation density conditioned on class $h$ and $k_{i-1}$ is the number of classes created at time $i-1$. The algorithm sequentially allocates observations $\mathbf{y}_i$ to classes based on maximizing the conditional posterior probability.

To calculate the posterior probability $P(\gamma_i = h|\mathbf{y}^{(i)}, \gamma^{(i-1)})$, define the variables:

$$L_{i,h}(\mathbf{y}_i) \stackrel{\text{def}}{=} P(\mathbf{y}_i|\gamma_i = h, \mathbf{y}^{(i-1)}, \gamma^{(i-1)}), \qquad \pi_{i,h}(\alpha) \stackrel{\text{def}}{=} P(\gamma_i = h|\alpha, \mathbf{y}^{(i-1)}, \gamma^{(i-1)})$$

From Bayes' rule, $P(\gamma_i = h|\mathbf{y}^{(i)}, \gamma^{(i-1)}) \propto L_{i,h}(\mathbf{y}_i)\pi_{i,h}(\alpha)$ for $h = 1, \ldots, k_{i-1}+1$. Here, $\alpha$ is considered fixed at this iteration, and is not updated in a fully Bayesian manner.

According to the Dirichlet process prediction, the predictive probability of assigning observation $\mathbf{y}_i$ to a class $h$ is:

$$\pi_{i,h}(\alpha) = \begin{cases} \frac{m_{i-1}(h)}{i-1+\alpha}, & h = 1, \ldots, k_{i-1} \\ \frac{\alpha}{i-1+\alpha}, & h = k_{i-1}+1 \end{cases} \tag{1}$$

---

**Algorithm 1** Adaptive Sequential Updating and Greedy Search (ASUGS)

---

**Input:** streaming data $\{\mathbf{y}_i\}_{i=1}^{\infty}$, rate parameter $\lambda > 0$.
Set $\gamma_1 = 1$ and $k_1 = 1$. Calculate $\pi(\theta_1 | \mathbf{y}_1, \gamma_1)$.
**for** $i \geq 2$: **do**

    (a) Update concentration parameter:    $\alpha_{i-1} = \frac{k_{i-1}}{\lambda + \log(i-1)}$.

    (b) Choose best label for $\mathbf{y}_i$:    $\gamma_i \sim \{q_h^{(i)}\} = \left\{ \frac{L_{i,h}(\mathbf{y}_i)\pi_{i,h}(\alpha_{i-1})}{\sum_{h'} L_{i,h'}(\mathbf{y}_i)\pi_{i,h'}(\alpha_{i-1})} \right\}$.

    (c) Update posterior distribution:    $\pi(\theta_{\gamma_i} | \mathbf{y}^{(i)}, \gamma^{(i)}) \propto f(\mathbf{y}_i | \theta_{\gamma_i})\pi(\theta_{\gamma_i} | \mathbf{y}^{(i-1)}, \gamma^{(i-1)})$.

**end for**

---

where $m_{i-1}(h) = \sum_{l=1}^{i-1} I(\gamma_l = h)$ counts the number of observations labeled as class $h$ at time $i - 1$, and $\alpha > 0$ is the concentration parameter.

## 2.1 Adaptation of Concentration Parameter $\alpha$

It is well known that the concentration parameter $\alpha$ has a strong influence on the growth of the number of classes Antoniak (1974). Our experiments show that in this sequential framework, the choice of $\alpha$ is even more critical. Choosing a fixed $\alpha$ as in the online SVA algorithm of Lin (2013) requires cross-validation, which is computationally prohibitive for large-scale data sets. Furthermore, in the streaming data setting where no estimate on the data complexity exists, it is impractical to perform cross-validation. Although the parameter $\alpha$ is handled from a fully Bayesian treatment in Wang & Dunson (2011), a pre-specified grid of possible values $\alpha$ can take, say $\{\alpha_l\}_{l=1}^{L}$, along with the prior distribution over them, needs to be chosen in advance. Storage and updating of a matrix of size $(k_{i-1} + 1) \times L$ and further marginalization is needed to compute $P(\gamma_i = h | \mathbf{y}^{(i)}, \gamma^{(i-1)})$ at each iteration $i$. Thus, we propose an alternative data-driven method for choosing $\alpha$ that works well in practice, is simple to compute and has theoretical guarantees.

The idea is to start with a prior distribution on $\alpha$ that favors small $\alpha$ and shape it into a posterior distribution using the data. Define $p_i(\alpha) = p(\alpha | \mathbf{y}^{(i)}, \gamma^{(i)})$ as the posterior distribution formed at time $i$, which will be used in ASUGS at time $i + 1$. Let $p_1(\alpha) \equiv p_1(\alpha | \mathbf{y}^{(1)}, \gamma^{(1)})$ denote the prior for $\alpha$, e.g., an exponential distribution $p_1(\alpha) = \lambda e^{-\lambda \alpha}$. The dependence on $\mathbf{y}^{(i)}$ and $\gamma^{(i)}$ is trivial only at this first step. Then, by Bayes rule, $p_i(\alpha) \propto p(\mathbf{y}_i, \gamma_i | \mathbf{y}^{(i-1)}, \gamma^{(i-1)}, \alpha)p(\alpha | \mathbf{y}^{(i-1)}, \gamma^{(i-1)}) \propto p_{i-1}(\alpha)\pi_{i,\gamma_i}(\alpha)$ where $\pi_{i,\gamma_i}(\alpha)$ is given in (1). Once this update is made after the selection of $\gamma_i$, the $\alpha$ to be used in the next selection step is the mean of the distribution $p_i(\alpha)$, i.e., $\alpha_i = \mathbb{E}[\alpha | \mathbf{y}^{(i)}, \gamma^{(i)}]$. As will be shown in Section 5, the distribution $p_i(\alpha)$ can be approximated by a Gamma distribution with shape parameter $k_i$ and rate parameter $\lambda + \log i$. Under this approximation, we have $\alpha_i = \frac{k_i}{\lambda + \log i}$, only requiring storage and update of one scalar parameter $k_i$ at each iteration $i$.

The ASUGS algorithm is summarized in Algorithm 1. The selection step may be implemented by sampling the probability mass function $\{q_h^{(i)}\}$. The posterior update step can be efficiently performed by updating the hyperparameters as a function of the streaming data for the case of conjugate distributions. Section 3 derives these updates for the case of multivariate Gaussian observations and conjugate priors for the parameters.

## 3 Sequential Inference under Unknown Mean & Unknown Covariance

We consider the general case of an unknown mean and covariance for each class. The probabilistic model for the parameters of each class is given as:

$$\mathbf{y}_i | \mu, \mathbf{T} \sim \mathcal{N}(\cdot | \mu, \mathbf{T}), \qquad \mu | \mathbf{T} \sim \mathcal{N}(\cdot | \mu_0, c_o \mathbf{T}), \qquad \mathbf{T} \sim \mathcal{W}(\cdot | \delta_0, \mathbf{V}_0) \tag{2}$$

where $\mathcal{N}(\cdot | \mu, \mathbf{T})$ denotes the multivariate normal distribution with mean $\mu$ and precision matrix $\mathbf{T}$, and $\mathcal{W}(\cdot | \delta, \mathbf{V})$ is the Wishart distribution with $2\delta$ degrees of freedom and scale matrix $\mathbf{V}$. The parameters $\theta = (\mu, \mathbf{T}) \in \mathbb{R}^d \times S_{++}^d$ follow a normal-Wishart joint distribution. The model (2) leads to closed-form expressions for $L_{i,h}(\mathbf{y}_i)$'s due to conjugacy Tzikas et al. (2008).

To calculate the class posteriors, the conditional likelihoods of $\mathbf{y}_i$ given assignment to class $h$ and the previous class assignments need to be calculated first. The conditional likelihood of $\mathbf{y}_i$ given

assignment to class $h$ and the history $(\mathbf{y}^{(i-1)}, \gamma^{(i-1)})$ is given by:

$$L_{i,h}(\mathbf{y}_i) = \int f(\mathbf{y}_i|\theta_h)\pi(\theta_h|\mathbf{y}^{(i-1)}, \gamma^{(i-1)})d\theta_h \qquad (3)$$

Due to the conjugacy of the distributions, the posterior $\pi(\theta_h|\mathbf{y}^{(i-1)}, \gamma^{(i-1)})$ always has the form:

$$\pi(\theta_h|\mathbf{y}^{(i-1)}, \gamma^{(i-1)}) = \mathcal{N}(\mu_h|\mu_h^{(i-1)}, c_h^{(i-1)}\mathbf{T}_h)\mathcal{W}(\mathbf{T}_h|\delta_h^{(i-1)}, \mathbf{V}_h^{(i-1)})$$

where $\mu_h^{(i-1)}, c_h^{(i-1)}, \delta_h^{(i-1)}, \mathbf{V}_h^{(i-1)}$ are hyperparameters that can be recursively computed as new samples come in. The form of this recursive computation of the hyperparameters is derived in Appendix A. For ease of interpretation and numerical stability, we define $\mathbf{\Sigma}_h^{(i)} := \frac{(\mathbf{V}_h^{(i)})^{-1}}{2\delta_h^{(i)}}$ as the inverse of the mean of the Wishart distribution $\mathcal{W}(\cdot|\delta_h^{(i)}, \mathbf{V}_h^{(i)})$. The matrix $\mathbf{\Sigma}_h^{(i)}$ has the natural interpretation as the covariance matrix of class $h$ at iteration $i$. Once the $\gamma_i$th component is chosen, the parameter updates for the $\gamma_i$th class become:

$$\mu_{\gamma_i}^{(i)} = \frac{1}{1 + c_{\gamma_i}^{(i-1)}}\mathbf{y}_i + \frac{c_{\gamma_i}^{(i-1)}}{1 + c_{\gamma_i}^{(i-1)}}\mu_{\gamma_i}^{(i-1)} \qquad (4)$$

$$c_{\gamma_i}^{(i)} = c_{\gamma_i}^{(i-1)} + 1 \qquad (5)$$

$$\mathbf{\Sigma}_{\gamma_i}^{(i)} = \frac{2\delta_{\gamma_i}^{(i-1)}}{1 + 2\delta_{\gamma_i}^{(i-1)}}\mathbf{\Sigma}_{\gamma_i}^{(i-1)} + \frac{1}{1 + 2\delta_{\gamma_i}^{(i-1)}}\frac{c_{\gamma_i}^{(i-1)}}{1 + c_{\gamma_i}^{(i-1)}}(\mathbf{y}_i - \mu_{\gamma_i}^{(i-1)})(\mathbf{y}_i - \mu_{\gamma_i}^{(i-1)})^T \qquad (6)$$

$$\delta_{\gamma_i}^{(i)} = \delta_{\gamma_i}^{(i-1)} + \frac{1}{2} \qquad (7)$$

If the starting matrix $\mathbf{\Sigma}_h^{(0)}$ is positive definite, then all the matrices $\{\mathbf{\Sigma}_h^{(i)}\}$ will remain positive definite. Let us return to the calculation of the conditional likelihood (3). By iterated integration, it follows that:

$$L_{i,h}(\mathbf{y}_i) \propto \left(\frac{r_h^{(i-1)}}{2\delta_h^{(i-1)}}\right)^{d/2} \frac{\rho_d(\delta_h^{(i-1)})\det(\mathbf{\Sigma}_h^{(i-1)})^{-1/2}}{\left(1 + \frac{r_h^{(i-1)}}{2\delta_h^{(i-1)}}(\mathbf{y}_i - \mu_h^{(i-1)})^T(\mathbf{\Sigma}_h^{(i-1)})^{-1}(\mathbf{y}_i - \mu_h^{(i-1)})\right)^{\delta_h^{(i-1)} + \frac{1}{2}}} \qquad (8)$$

where $\rho_d(a) \overset{\text{def}}{=} \frac{\Gamma(a+\frac{1}{2})}{\Gamma(a+\frac{1-d}{2})}$ and $r_h^{(i-1)} \overset{\text{def}}{=} \frac{c_h^{(i-1)}}{1+c_h^{(i-1)}}$. A detailed mathematical derivation of this conditional likelihood is included in Appendix B. We remark that for the new class $h = k_{i-1} + 1$, $L_{i,k_{i-1}+1}$ has the form (8) with the initial choice of hyperparameters $r^{(0)}, \delta^{(0)}, \mu^{(0)}, \mathbf{\Sigma}^{(0)}$.

## 4 Growth Rate Analysis of Number of Classes & Stability

In this section, we derive a model for the posterior distribution $p_n(\alpha)$ using large-sample approximations, which will allow us to derive growth rates on the number of classes and the sequence of concentration parameters, showing that the number of classes grows as $\mathbb{E}[k_n] = O(\log^{1+\epsilon} n)$ for $\epsilon$ arbitarily small under certain mild conditions.

The probability density of the $\alpha$ parameter is updated at the $j$th step in the following fashion:

$$p_{j+1}(\alpha) \propto p_j(\alpha) \cdot \begin{cases} \frac{\alpha}{j+\alpha} & \text{innovation class chosen} \\ \frac{1}{j+\alpha} & \text{otherwise} \end{cases},$$

where only the $\alpha$-dependent factors in the update are shown. The $\alpha$-independent factors are absorbed by the normalization to a probability density. Choosing the innovation class pushes mass toward infinity while choosing any other class pushes mass toward zero. Thus there is a possibility that the innovation probability grows in a undesired manner. We assess the growth of the number of innovations $r_n \overset{\text{def}}{=} k_n - 1$ under simple assumptions on some likelihood functions that appear naturally in the ASUGS algorithm.

Assuming that the initial distribution of $\alpha$ is $p_1(\alpha) = \lambda e^{-\lambda\alpha}$, the distribution used at step $n + 1$ is proportional to $\alpha^{r_n}\prod_{j=1}^{n-1}(1 + \frac{\alpha}{j})^{-1}e^{-\lambda\alpha}$. We make use of the limiting relation

**Theorem 1.** *The following asymptotic behavior holds:* $\lim_{n \to \infty} \frac{\log \prod_{j=1}^{n-1} (1+\frac{\alpha}{j})}{\alpha \log n} = 1$.

*Proof.* See Appendix C. ☐

Using Theorem 1, a large-sample model for $p_n(\alpha)$ is $\alpha^{r_n} e^{-(\lambda + \log n)\alpha}$, suitably normalized. Recognizing this as the Gamma distribution with shape parameter $r_n + 1$ and rate parameter $\lambda + \log n$, its mean is given by $\alpha_n = \frac{r_n + 1}{\lambda + \log n}$. We use the mean in this form to choose class membership in Alg. 1. This asymptotic approximation leads to a very simple scalar update of the concentration parameter; there is no need for discretization for tracking the evolution of continuous probability distributions on $\alpha$. In our experiments, this approximation is very accurate.

Recall that the innovation class is labeled $K_+ = k_{n-1} + 1$ at the $n^{th}$ step. The modeled updates randomly select a previous class or innovation (new class) by sampling from the probability distribution $\{q_k^{(n)} = P(\gamma_n = k | \mathbf{y}^{(n)}, \gamma^{(n-1)})\}_{k=1}^{K_+}$. Note that $n - 1 = \sum_{k \neq K_+} m_n(k)$, where $m_n(k)$ represents the number of members in class $k$ at time $n$.

We assume the data follows the Gaussian mixture distribution:

$$p_T(\mathbf{y}) \overset{\text{def}}{=} \sum_{h=1}^{K} \pi_h \mathcal{N}(\mathbf{y}|\mu_h, \mathbf{\Sigma}_h) \tag{9}$$

where $\pi_h$ are the prior probabilities, and $\mu_h, \mathbf{\Sigma}_h$ are the parameters of the Gaussian clusters.

Define the mixture-model probability density function, which plays the role of the predictive distribution:

$$\tilde{L}_{n,K_+}(\mathbf{y}) \overset{\text{def}}{=} \sum_{k \neq K_+} \frac{m_{n-1}(k)}{n - 1} L_{n,k}(\mathbf{y}), \tag{10}$$

so that the probabilities of choosing a previous class or an innovation (using Equ. (1)) are proportional to $\sum_{k \neq K_+} \frac{m_{n-1}(k)}{n-1+\alpha_{n-1}} L_{n,k}(\mathbf{y}_n) = \frac{(n-1)}{n-1+\alpha_{n-1}} \tilde{L}_{n,K_+}(\mathbf{y}_n)$ and $\frac{\alpha_{n-1}}{n-1+\alpha_{n-1}} L_{n,K_+}(\mathbf{y}_n)$, respectively. If $\tau_{n-1}$ denotes the innovation probability at step $n$, then we have

$$\left( \rho_{n-1} \frac{\alpha_{n-1} L_{n,K_+}(\mathbf{y}_n)}{n - 1 + \alpha_{n-1}}, \rho_{n-1} \frac{(n-1)\tilde{L}_{n,K_+}(\mathbf{y}_n)}{n - 1 + \alpha_{n-1}} \right) = (\tau_{n-1}, 1 - \tau_{n-1}) \tag{11}$$

for some positive proportionality factor $\rho_{n-1}$.

Define the likelihood ratio (LR) at the beginning of stage $n$ as [1]:

$$l_n(\mathbf{y}) \overset{\text{def}}{=} \frac{L_{n,K_+}(\mathbf{y})}{\tilde{L}_{n,K_+}(\mathbf{y})} \tag{12}$$

Conceptually, the mixture (10) represents a modeled distribution fitting the currently observed data. If all "modes" of the data have been observed, it is reasonable to expect that $\tilde{L}_{n,K_+}$ is a good model for future observations. The LR $l_n(\mathbf{y}_n)$ is not large when the future observations are well-modeled by (10). In fact, we expect $\tilde{L}_{n,K_+} \to p_T$ as $n \to \infty$, as discussed in Section 5.

**Lemma 1.** *The following bound holds:* $\tau_{n-1} = \frac{l_n(\mathbf{y}_n)\alpha_{n-1}}{n-1+l_n(\mathbf{y}_n)\alpha_{n-1}} \leq \min\left( \frac{l_n(\mathbf{y}_n)\alpha_{n-1}}{n-1}, 1 \right)$.

*Proof.* The result follows directly from (11) after a simple calculation. ☐

The innovation random variable $r_n$ is described by the random process associated with the probabilities of transition

$$P(r_{n+1} = k | r_n) = \begin{cases} \tau_n, & k = r_n + 1 \\ 1 - \tau_n, & k = r_n \end{cases}. \tag{13}$$

The expectation of $r_n$ is majorized by the expectation of a similar random process, $\bar{r}_n$, based on the transition probability $\sigma_n \overset{\text{def}}{=} \min(\frac{r_n+1}{a_n}, 1)$ instead of $\tau_n$ as Appendix D shows, where the random sequence $\{a_n\}$ is given by $l_{n+1}(\mathbf{y}_{n+1})^{-1}n(\lambda+\log n)$. The latter can be described as a modification of a Polya urn process with selection probability $\sigma_n$. The asymptotic behavior of $r_n$ and related variables is described in the following theorem.

**Theorem 2.** *Let $\tau_n$ be a sequence of real-valued random variables $0 \leq \tau_n \leq 1$ satisfying $\tau_n \leq \frac{r_n+1}{a_n}$ for $n \geq N$, where $a_n = l_{n+1}(\mathbf{y}_{n+1})^{-1}n(\lambda + \log n)$, and where the nonnegative, integer-valued random variables $r_n$ evolve according to (13). Assume the following for $n \geq N$:*

    *1. $l_n(\mathbf{y}_n) \leq \zeta$   (a.s.)*

    *2. $D(p_T \parallel \tilde{L}_{n,K+}) \leq \delta$   (a.s.)*

*where $D(p \parallel q)$ is the Kullback-Leibler divergence between distributions $p(\cdot)$ and $q(\cdot)$. Then, as $n \to \infty$,*

$$r_n = O_P(\log^{1+\zeta\sqrt{\delta/2}} n), \qquad \alpha_n = O_P(\log^{\zeta\sqrt{\delta/2}} n) \tag{14}$$

*Proof.* See Appendix E. $\qquad\qquad\qquad\qquad\qquad\qquad\qquad\qquad\qquad\qquad\qquad\qquad\qquad\qquad\square$

Theorem 2 bounds the growth rate of the mean of the number of class innovations and the concentration parameter $\alpha_n$ in terms of the sample size $n$ and parameter $\zeta$. The bounded LR and bounded KL divergence conditions of Thm. 2 manifest themselves in the rate exponents of (14). The experiments section shows that both of the conditions of Thm. 2 hold for all iterations $n \geq N$ for some $N \in \mathbb{N}$. In fact, assuming the correct clustering, the mixture distribution $\tilde{L}_{n,k_{n-1}+1}$ converges to the true mixture distribution $p_T$, implying that the number of class innovations grows at most as $O(\log^{1+\epsilon} n)$ and the sequence of concentration parameters is $O(\log^\epsilon n)$, where $\epsilon > 0$ can be arbitrarily small.

## 5 Asymptotic Normality of Conditional Likelihood

In this section, we derive an asymptotic expression for the conditional likelihood (8) in order to gain insight into the steady-state of the algorithm.

We let $\pi_h$ denote the true prior probability of class $h$. Using the bounds of the Gamma function in Theorem 1.6 from Batir (2008), it follows that $\lim_{a\to\infty} \frac{\rho_d(a)}{e^{-d/2}(a-1/2)^{d/2}} = 1$. Under normal convergence conditions of the algorithm (with the pruning and merging steps included), all classes $h = 1, \ldots, K$ will be correctly identified and populated with approximately $n_{i-1}(h) \approx \pi_h(i-1)$ observations at time $i-1$. Thus, the conditional class prior for each class $h$ converges to $\pi_h$ as $i \to \infty$, in virtue of (14), $\pi_{i,h}(\alpha_{i-1}) = \frac{n_{i-1}(h)}{i-1+\alpha_{i-1}} = \frac{\pi_h}{1+\frac{O_P(\log^\zeta\sqrt{\delta/2}(i-1))}{i-1}} \overset{i\to\infty}{\longrightarrow} \pi_h$. According to (5), we expect $r_h^{(i-1)} \to 1$ as $i \to \infty$ since $c_h^{(i-1)} \sim \pi_h(i-1)$. Also, we expect $2\delta_h^{(i-1)} \sim \pi_h(i-1)$ as $i \to \infty$ according to (7). Also, from before, $\rho_d(\delta_h^{(i-1)}) \sim e^{-d/2}(\delta_h^{(i-1)} - 1/2)^{d/2} \sim e^{-d/2}(\pi_h\frac{i-1}{2} - \frac{1}{2})^{d/2}$. The parameter updates (4)-(7) imply $\mu_h^{(i)} \to \mu_h$ and $\mathbf{\Sigma}_h^{(i)} \to \mathbf{\Sigma}_h$ as $i \to \infty$. This follows from the strong law of large numbers, as the updates are recursive implementations of the sample mean and sample covariance matrix. Thus, the large-sample approximation to the conditional likelihood becomes:

$$L_{i,h}(\mathbf{y}_i) \overset{i\to\infty}{\propto} \frac{\lim_{i\to\infty}\left(1 + \frac{\pi_h^{-1}}{i-1}(\mathbf{y}_i - \mu_h^{(i-1)})^T(\mathbf{\Sigma}_h^{(i-1)})^{-1}(\mathbf{y}_i - \mu_h^{(i-1)})\right)^{-\frac{i-1}{2\pi_h^{-1}}}}{\lim_{i\to\infty}\det(\mathbf{\Sigma}_h^{(i-1)})^{1/2}}$$

$$\overset{i\to\infty}{\propto} \frac{e^{-\frac{1}{2}(\mathbf{y}_i - \mu_h)^T\mathbf{\Sigma}_h^{-1}(\mathbf{y}_i - \mu_h)}}{\sqrt{\det\mathbf{\Sigma}_h}} \tag{15}$$

where we used $\lim_{u\to\infty}(1+\frac{c}{u})^u = e^c$. The conditional likelihood (15) corresponds to the multivariate Gaussian distribution with mean $\mu_h$ and covariance matrix $\mathbf{\Sigma}_h$. A similar asymptotic normality

result was recently obtained in Tsiligkaridis & Forsythe (2015) for Gaussian observations with a von Mises prior. The asymptotics $\frac{m_{n-1}(h)}{n-1} \to \pi_h$, $\mu_h^{(n)} \to \mu_h$, $\boldsymbol{\Sigma}_h^{(n)} \to \boldsymbol{\Sigma}_h$, $L_{n,h}(\mathbf{y}) \to \mathcal{N}(\mathbf{y}|\mu_h, \boldsymbol{\Sigma}_h)$ as $n \to \infty$ imply that the mixture distribution $\tilde{L}_{n,K+}$ in (10) converges to the true Gaussian mixture distribution $p_T$ of (9). Thus, for any small $\delta$, we expect $D(p_T \parallel \tilde{L}_{n,K+}) \le \delta$ for all $n \ge N$, validating the assumption of Theorem 2.

# 6 Experiments

We apply the ASUGS learning algorithm to a synthetic 16-class example and to a real data set, to verify the stability and accuracy of our method. The experiments show the value of adaptation of the Dirichlet concentration parameter for online clustering and parameter estimation.

Since it is possible that multiple clusters are similar and classes might be created due to outliers, or due to the particular ordering of the streaming data sequence, we add the pruning and merging step in the ASUGS algorithm as done in Lin (2013). We compare ASUGS and ASUGS-PM with SUGS, SUGS-PM, SVA and SVA-PM proposed in Lin (2013), since it was shown in Lin (2013) that SVA and SVA-PM outperform the block-based methods that perform iterative updates over the entire data set including Collapsed Gibbs Sampling, MCMC with Split-Merge and Truncation-Free Variational Inference.

## 6.1 Synthetic Data set

We consider learning the parameters of a 16-class Gaussian mixture each with equal variance of $\sigma^2 = 0.025$. The training set was made up of 500 iid samples, and the test set was made up of 1000 iid samples. The clustering results are shown in Fig. 1(a), showing that the ASUGS-based approaches are more stable than SVA-based algorithms. ASUGS-PM performs best and identifies the correct number of clusters, and their parameters. Fig. 1(b) shows the data log-likelihood on the test set (averaged over 100 Monte Carlo trials), the mean and variance of the number of classes at each iteration. The ASUGS-based approaches achieve a higher log-likelihood than SVA-based approaches asymptotically. Fig. 6.1 provides some numerical verification for the assumptions of Theorem 2. As expected, the predictive likelihood $\tilde{L}_{i,K+}$ (10) converges to the true mixture distribution $p_T$ (9), and the likelihood ratio $l_i(\mathbf{y}_i)$ is bounded after enough samples are processed.

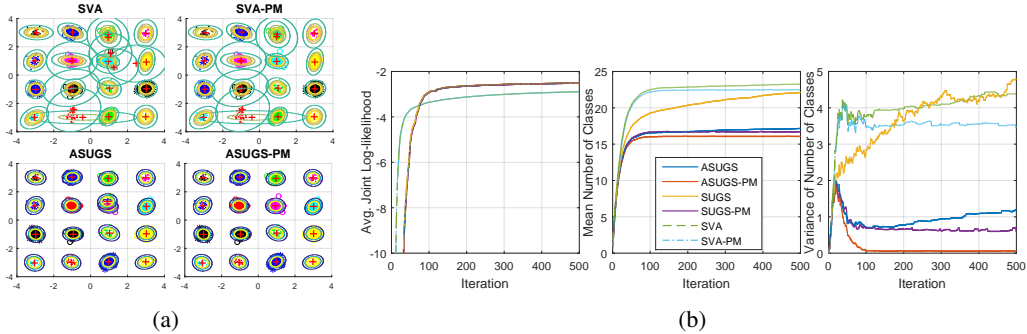

Figure 1: (a) Clustering performance of SVA, SVA-PM, ASUGS and ASUGS-PM on synthetic data set. ASUGS-PM identifies the 16 clusters correctly. (b) Joint log-likelihood on synthetic data, mean and variance of number of classes as a function of iteration. The likelihood values were evaluated on a held-out set of 1000 samples. ASUGS-PM achieves the highest log-likelihood and has the lowest asymptotic variance on the number of classes.

## 6.2 Real Data Set

We applied the online nonparametric Bayesian methods for clustering image data. We used the MNIST data set, which consists of $60,000$ training samples, and $10,000$ test samples. Each sample

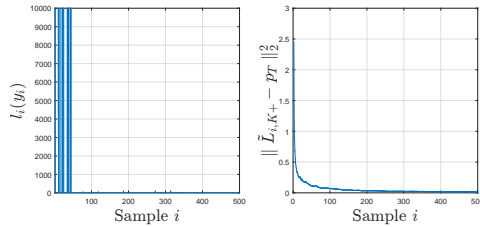

Figure 2: Likelihood ratio $l_i(\mathbf{y}_i) = \frac{L_{i,K+}(\mathbf{y}_i)}{\tilde{L}_{i,K+}(\mathbf{y}_i)}$ (left) and $L_2$-distance between $\tilde{L}_{i,K+}(\cdot)$ and true mixture distribution $p_T$ (right) for synthetic example (see 1).

is a $28 \times 28$ image of a handwritten digit (total of $784$ dimensions), and we perform PCA pre-processing to reduce dimensionality to $d = 50$ dimensions as in Kurihara et al. (2006).

We use only a random $1.667\%$ subset, consisting of $1000$ random samples for training. This training set contains data from all $10$ digits with an approximately uniform proportion. Fig. 3 shows the predictive log-likelihood over the test set, and the mean images for clusters obtained using ASUGS-PM and SVA-PM, respectively. We note that ASUGS-PM achieves higher log-likelihood values and finds all digits correctly using only $23$ clusters, while SVA-PM finds some digits using $56$ clusters.

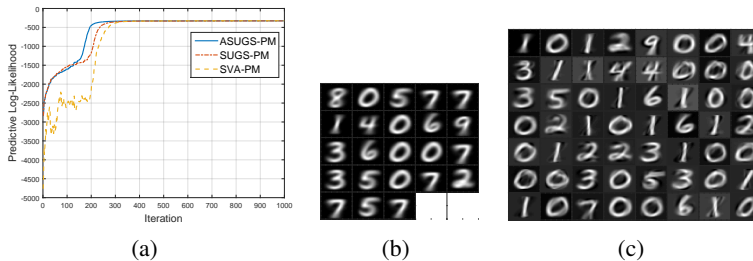

(a)                     (b)                     (c)

Figure 3: Predictive log-likelihood (a) on test set, mean images for clusters found using ASUGS-PM (b) and SVA-PM (c) on MNIST data set.

## 6.3  Discussion

Although both SVA and ASUGS methods have similar computational complexity and use decisions and information obtained from processing previous samples in order to decide on class innovations, the mechanics of these methods are quite different. ASUGS uses an adaptive $\alpha$ motivated by asymptotic theory, while SVA uses a fixed $\alpha$. Furthermore, SVA updates the parameters of all the components at each iteration (in a weighted fashion) while ASUGS only updates the parameters of the most-likely cluster, thus minimizing leakage to unrelated components. The $\lambda$ parameter of ASUGS does not affect performance as much as the threshold parameter $\epsilon$ of SVA does, which often leads to instability requiring lots of pruning and merging steps and increasing latency. This is critical for large data sets or streaming applications, because cross-validation would be required to set $\epsilon$ appropriately. We observe higher log-likelihoods and better numerical stability for ASUGS-based methods in comparison to SVA. The mathematical formulation of ASUGS allows for theoretical guarantees (Theorem 2), and asymptotically normal predictive distribution.

## 7  Conclusion

We developed a fast online clustering and parameter estimation algorithm for Dirichlet process mixtures of Gaussians, capable of learning in a single data pass. Motivated by large-sample asymptotics, we proposed a novel low-complexity data-driven adaptive design for the concentration parameter and showed it leads to logarithmic growth rates on the number of classes. Through experiments on synthetic and real data sets, we show our method achieves better performance and is as fast as other state-of-the-art online learning DPMM methods.

## Footnotes

[1]Here, $L_0(\cdot) \overset{\text{def}}{=} L_{n,K_+}(\cdot)$ is independent of $n$ and only depends on the initial choice of hyperparameters as discussed in Sec. 3.

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
