[Supplementary Material]

# Adaptive Low-Complexity Sequential Inference for Dirichlet Process Mixture Models

**Theodoros Tsiligkaridis,    Keith W. Forsythe**
Massachusetts Institute of Technology, Lincoln Laboratory
Lexington, MA 02421 USA
ttsili@ll.mit.edu, forsythe@ll.mit.edu

## 1  Appendix A

We consider the general case of an unknown mean and covariance for each class. Let $\mathbf{T}$ denote the precision (or inverse covariance) matrix. The probabilistic model for the mean and covariance matrix of each class is given as:

$$
\begin{aligned}
\mathbf{y}_i|\mu,\mathbf{T} &\sim \mathcal{N}(\cdot|\mu,\mathbf{T})\\
\mu|\mathbf{T} &\sim \mathcal{N}(\cdot|\mu_0,c_o\mathbf{T})\\
\mathbf{T} &\sim \mathcal{W}(\cdot|\delta_0,\mathbf{V}_0)
\end{aligned}
\tag{1}
$$

where $\mathcal{N}(\cdot|\mu,\mathbf{T})$ denote the observation density which is assumed to be multivariate normal with mean $\mu$ and precision matrix $\mathbf{T}$. The parameters $\theta = (\mu,\mathbf{T}) \in \Omega_1 \times \Omega_2$ follow a normal-Wishart joint distribution. The domains here are $\Omega_1 = \mathbb{R}^d$ and $\Omega_2 = S^d_{++}$ is the positive definite cone. This leads to closed-form expressions for $L_{i,h}(\mathbf{y}_i)$'s due to conjugacy Tzikas et al. (2008). For concreteness, let us write the distributions of the model (1):

$$
f(\mathbf{y}_i|\theta) = p(\mathbf{y}_i|\mu,\mathbf{T}) = \frac{\det(\mathbf{T})^{1/2}}{(2\pi)^{d/2}}\exp\left(-\frac{1}{2}(\mathbf{y}_i-\mu)^T\mathbf{T}(\mathbf{y}_i-\mu)\right)
$$

$$
p(\mu|\mathbf{T}) = p(\theta_1|\mathbf{\Theta}_2) = \frac{\det(c_0\mathbf{T})^{1/2}}{(2\pi)^{d/2}}\exp\left(-\frac{c_0}{2}(\mu-\mu_0)^T\mathbf{T}(\mu-\mu_0)\right)
$$

$$
p(\mathbf{T}) = p(\mathbf{\Theta}_2) = \frac{\det(\mathbf{V}_0)^{-\delta_0}}{2^{d\delta_0}\Gamma_d(\delta_0)}\det(\mathbf{T})^{\delta_0-\frac{d+1}{2}}\exp(-\frac{1}{2}\mathrm{tr}(\mathbf{V}_0^{-1}\mathbf{T}))
$$

where $\Gamma_d(\cdot)$ is the multivariate Gamma function.

To calculate the class posteriors, the conditional likelihoods of $\mathbf{y}_i$ given assignment to class $h$ and the previous class assignments need to be calculated first. We derive closed-form expressions for these quantities in this section under the probabilistic model (1).

The conditional likelihood of $\mathbf{y}_i$ given assignment to class $h$ and the history $(\mathbf{y}^{(i-1)},\gamma^{(i-1)})$ is given by:

$$
L_{i,h}(\mathbf{y}_i) = \int f(\mathbf{y}_i|\theta_h)\pi(\theta_h|\mathbf{y}^{(i-1)},\gamma^{(i-1)})d\theta_h
\tag{2}
$$

We thus need to obtain an expression for the posterior distribution $\pi(\theta_h|\mathbf{y}^{(i-1)},\gamma^{(i-1)})$. Due to the conjugacy of the distributions involved in (1), the posterior distribution $\pi(\theta_h|\mathbf{y}^{(i-1)},\gamma^{(i-1)})$ always has the form:

$$
\pi(\theta_h|\mathbf{y}^{(i-1)},\gamma^{(i-1)}) = \mathcal{N}(\mu_h|\mu_h^{(i-1)},c_h^{(i-1)}\mathbf{T}_h)\mathcal{W}(\mathbf{T}_h|\delta_h^{(i-1)},\mathbf{V}_h^{(i-1)})
\tag{3}
$$

where $\mu_h^{(i-1)},c_h^{(i-1)},\delta_h^{(i-1)},\mathbf{V}_h^{(i-1)}$ are hyperparameters that can be recursively computed as new samples come in. This would greatly simplify the computational complexity of the second step of the SUGS algorithm. Next, we derive the form of this recursive computation of the hyperparameters.

For simplicity of the derivation, let us consider the initial case $\mathbf{y} = \mathbf{y}_1$. Then, from Bayes' rule:

$$p(\theta|\mathbf{y}) = p(\mu, \mathbf{T}|\mathbf{y}) = p(\mu|\mathbf{T}, \mathbf{y})p(\mathbf{T}|\mathbf{y})$$

## 1.1 Calculation of $p(\mu|\mathbf{T}, \mathbf{y})$

Note the factorization:

$$p(\mu|\mathbf{T}, \mathbf{y}) \propto p(\mathbf{y}|\mu, \mathbf{T})p(\mu|\mathbf{T})$$

According to (1), we can write:

$$\mathbf{y} = \mu + \boldsymbol{\Sigma}^{1/2}\epsilon$$

$$\mu = \mu_0 + \boldsymbol{\Sigma}_0^{1/2}\epsilon'$$

where $\epsilon \sim N(0, I), \epsilon' \sim N(0, I)$, $\epsilon$ is independent of $\epsilon'$ and $\boldsymbol{\Sigma} = \mathbf{T}^{-1}, \boldsymbol{\Sigma}_0 = (c_0\mathbf{T})^{-1}$. From this, it follows that the conditional density $p(\mu|\mathbf{T}, \mathbf{y})$ is also multivariate normal with mean $\mathbb{E}[\mu|\mathbf{T}, \mathbf{y}]$ and covariance $\text{Cov}(\mu|\mathbf{T}, \mathbf{y})$. Note that:

$$\mathbb{E}[\mathbf{y}|\mathbf{T}] = \mu_0$$
$$\text{Cov}(\mathbf{y}|\mathbf{T}) = \mathbb{E}[\text{Cov}(\mathbf{y}|\mu, \mathbf{T})|\mathbf{T}] + \text{Cov}(\mathbb{E}[\mathbf{y}|\mu, \mathbf{T}]|\mathbf{T})$$
$$= \boldsymbol{\Sigma} + \boldsymbol{\Sigma}_0 = (1 + c_0^{-1})\mathbf{T}^{-1}$$
$$\text{Cov}(\mu, \mathbf{y}|\mathbf{T}) = \boldsymbol{\Sigma}_0$$

Using these facts, we obtain:

$$\mathbb{E}[\mu|\mathbf{T}, \mathbf{y}] = \mathbb{E}[\mu|\mathbf{T}] + \text{Cov}(\mu, \mathbf{y}|\mathbf{T})\text{Cov}(\mathbf{y}|\mathbf{T})^{-1}(\mathbf{y} - \mathbb{E}[\mathbf{y}|\mathbf{T}])$$
$$= \mu_0 + c_0^{-1}\mathbf{T}^{-1}((1 + c_0^{-1})\mathbf{T}^{-1})^{-1}(\mathbf{y} - \mu_0)$$
$$= \mu_0 + c_0^{-1}(1 + c_0^{-1})^{-1}(\mathbf{y} - \mu_0)$$
$$= \frac{1}{1 + c_0}\mathbf{y} + \frac{c_0}{1 + c_0}\mu_0$$
$$\text{Cov}(\mu|\mathbf{T}, \mathbf{y}) = \text{Cov}(\mu|\mathbf{T}) - \text{Cov}(\mu, \mathbf{y}|\mathbf{T})\text{Cov}(\mathbf{y}|\mathbf{T})^{-1}\text{Cov}(\mathbf{y}, \mu|\mathbf{T})$$
$$= \boldsymbol{\Sigma}_0 - \boldsymbol{\Sigma}_0(\boldsymbol{\Sigma} + \boldsymbol{\Sigma}_0)^{-1}\boldsymbol{\Sigma}_0^T$$
$$= c_0^{-1}\left(1 - \frac{c_0^{-1}}{1 + c_0^{-1}}\right)\mathbf{T}^{-1}$$
$$= \frac{c_0^{-1}}{1 + c_0^{-1}}\mathbf{T}^{-1}$$

Thus, we have:

$$p(\mu|\mathbf{T}, \mathbf{y}) = \mathcal{N}\left(\mu \left| \frac{1}{1 + c_0}\mathbf{y} + \frac{c_0}{1 + c_0}\mu_0, (1 + c_0)\mathbf{T}\right.\right)$$

where the conditional precision matrix becomes $(1 + c_0)\mathbf{T}$. As a result, once the $\gamma_i$th component is chosen in the SUGS selection step, the parameter updates for the $\gamma_i$th class become:

$$\mu_{\gamma_i}^{(i)} = \frac{1}{1 + c_{\gamma_i}^{(i-1)}}\mathbf{y}_i + \frac{c_{\gamma_i}^{(i-1)}}{1 + c_{\gamma_i}^{(i-1)}}\mu_{\gamma_i}^{(i-1)}$$
$$c_{\gamma_i}^{(i)} = c_{\gamma_i}^{(i-1)} + 1 \tag{4}$$

## 1.2 Calculation of $p(\mathbf{T}|\mathbf{y})$

Next, we focus on calculating $p(\mathbf{T}|\mathbf{y}) = \int_{\mathbb{R}^d} p(\mathbf{T}, \mu|\mathbf{y})d\mu$, where

$$p(\mathbf{T}, \mu|\mathbf{y}) \propto p(\mathbf{y}|\mathbf{T}, \mu)p(\mu|\mathbf{T})p(\mathbf{T})$$
$$\propto \det(\mathbf{T})^{(\delta_0 + 1/2) - \frac{d+1}{2}}\det(\mathbf{T})^{1/2}\exp\left(-\frac{1}{2}\text{tr}(\mathbf{V}_0^{-1}\mathbf{T})\right)$$
$$\times \exp\left(-\frac{1}{2}\left[c_0(\mu - \mu_0)^T\mathbf{T}(\mu - \mu_0) + (\mathbf{y} - \mu)^T\mathbf{T}(\mathbf{y} - \mu)\right]\right)$$

Rewriting the term inside the brackets by completing the square, we obtain:

$$
\begin{aligned}
c_0(\mu - \mu_0)^T & \mathbf{T}(\mu - \mu_0) + (\mathbf{y} - \mu)^T \mathbf{T}(\mathbf{y} - \mu) \\
&= c_0 \|\mathbf{T}^{1/2}\mu - \mathbf{T}^{1/2}\mu_0\|_2^2 + \|\mathbf{T}^{1/2}\mathbf{y} - \mathbf{T}^{1/2}\mu\|_2^2 \\
&= (1 + c_0) \left\{ \|\mathbf{T}^{1/2}\mu\|_2^2 - 2 \left\langle \mathbf{T}^{1/2}\mu, \frac{c_0 \mathbf{T}^{1/2}\mu_0 + \mathbf{T}^{1/2}\mathbf{y}}{1 + c_0} \right\rangle + \frac{c_0 \|\mathbf{T}^{1/2}\mu_0\|_2^2 + \|\mathbf{T}^{1/2}\mathbf{y}\|_2^2}{1 + c_0} \right\} \\
&= (1 + c_0) \left\{ \|\mathbf{T}^{1/2}\mu - \frac{c_0 \mathbf{T}^{1/2}\mu_0 + \mathbf{T}^{1/2}\mathbf{y}}{1 + c_0}\|_2^2 - \|\frac{c_0 \mathbf{T}^{1/2}\mu_0 + \mathbf{T}^{1/2}\mathbf{y}}{1 + c_0}\|_2^2 + \frac{c_0 \|\mathbf{T}^{1/2}\mu_0\|_2^2 + \|\mathbf{T}^{1/2}\mathbf{y}\|_2^2}{1 + c_0} \right\}
\end{aligned}
$$

Integrating out $\mu$, we obtain:

$$
\begin{aligned}
\int \exp & \left( -\frac{1}{2} \left[ c_0(\mu - \mu_0)^T \mathbf{T}(\mu - \mu_0) + (\mathbf{y} - \mu)^T \mathbf{T}(\mathbf{y} - \mu) \right] \right) d\mu \\
&= \exp \left( -\frac{1 + c_0}{2} \left( \frac{c_0 \|\mathbf{T}^{1/2}\mu_0\|_2^2 + \|\mathbf{T}^{1/2}\mathbf{y}\|_2^2}{1 + c_0} - \|\frac{c_0 \mathbf{T}^{1/2}\mu_0 + \mathbf{T}^{1/2}\mathbf{y}}{1 + c_0}\|_2^2 \right) \right) \\
&\quad \times \int \exp(-\frac{1}{2} \|\mathbf{T}^{1/2}\mu - \frac{c_0 \mathbf{T}^{1/2}\mu_0 + \mathbf{T}^{1/2}\mathbf{y}}{1 + c_0}\|_2^2) d\mu \\
&\propto \det(\mathbf{T})^{-1/2} \exp \left( -\frac{1}{2} \frac{c_0}{1 + c_0} (\mathbf{y} - \mu_0)^T \mathbf{T}(\mathbf{y} - \mu_0) \right)
\end{aligned}
$$

Using this result, we obtain:

$$
p(\mathbf{T}|\mathbf{y}) \propto \det(\mathbf{T})^{(\delta_0 + 1/2) - \frac{d+1}{2}} \exp \left( -\frac{1}{2} \mathrm{tr} \left( \mathbf{T} \left\{ \mathbf{V}_0^{-1} + \frac{c_0}{1 + c_0} (\mathbf{y} - \mu_0)(\mathbf{y} - \mu_0)^T \right\} \right) \right)
$$

As a result, the conditional density is recognized to be a Wishart distribution

$$
\mathcal{W} \left( \mathbf{T} \middle| \delta_0 + \frac{1}{2}, \left\{ \mathbf{V}_0^{-1} + \frac{c_0}{1 + c_0} (\mathbf{y} - \mu_0)(\mathbf{y} - \mu_0)^T \right\}^{-1} \right).
$$

Thus, the parameter updates for the $\gamma_i$th class become:

$$
\delta_{\gamma_i}^{(i)} = \delta_{\gamma_i}^{(i-1)} + \frac{1}{2}
$$

$$
\mathbf{V}_{\gamma_i}^{(i)} = \left\{ (\mathbf{V}_{\gamma_i}^{(i-1)})^{-1} + \frac{c_{\gamma_i}^{(i-1)}}{1 + c_{\gamma_i}^{(i-1)}} (\mathbf{y}_i - \mu_{\gamma_i}^{(i-1)})(\mathbf{y}_i - \mu_{\gamma_i}^{(i-1)})^T \right\}^{-1} \tag{5}
$$

For numerical stability and ease of interpretation, we define

$$
\boldsymbol{\Sigma}_h^{(i)} := \frac{(\mathbf{V}_h^{(i)})^{-1}}{2\delta_h^{(i)}}.
$$

This is the inverse of the mean of the Wishart distribution $\mathcal{W}(\cdot|\delta_h^{(i)}, \mathbf{V}_h^{(i)})$, and can be interpreted as the covariance matrix of class $h$ at iteration $i$. From (5), we have:

$$
\begin{aligned}
\boldsymbol{\Sigma}_h^{(i)} &= \frac{(\mathbf{V}_h^{(i)})^{-1}}{2\delta_h^{(i)}} \\
&= \frac{2\delta_h^{(i-1)}}{2\delta_h^{(i)}} \frac{(\mathbf{V}_h^{(i-1)})^{-1}}{2\delta_h^{(i-1)}} + \frac{1}{2\delta_h^{(i)}} \frac{c_h^{(i-1)}}{1 + c_h^{(i-1)}} (\mathbf{y}_i - \mu_h^{(i-1)})(\mathbf{y}_i - \mu_h^{(i-1)})^T \\
&= \frac{2\delta_h^{(i-1)}}{1 + 2\delta_h^{(i-1)}} \boldsymbol{\Sigma}_h^{(i-1)} + \frac{1}{1 + 2\delta_h^{(i-1)}} \frac{c_h^{(i-1)}}{1 + c_h^{(i-1)}} (\mathbf{y}_i - \mu_h^{(i-1)})(\mathbf{y}_i - \mu_h^{(i-1)})^T
\end{aligned}
$$

Thus, the recursive updates (5) can be equivalently restated as:

$$\delta_{\gamma_i}^{(i)} = \delta_{\gamma_i}^{(i-1)} + \frac{1}{2}$$

$$\mathbf{\Sigma}_h^{(i)} = \frac{2\delta_h^{(i-1)}}{1 + 2\delta_h^{(i-1)}} \mathbf{\Sigma}_h^{(i-1)} + \frac{1}{1 + 2\delta_h^{(i-1)}} \frac{c_h^{(i-1)}}{1 + c_h^{(i-1)}} (\mathbf{y}_i - \mu_h^{(i-1)})(\mathbf{y}_i - \mu_h^{(i-1)})^T \qquad (6)$$

If the starting matrix $\mathbf{\Sigma}_h^{(0)}$ is positive definite, then all the matrices $\{\mathbf{\Sigma}_h^{(i)}\}$ will remain positive definite.

## 2 Appendix B

Now, let us return to the calculation of (2).

$$L_{i,h}(\mathbf{y}_i) = \int_{S_{++}^d} \int_{\mathbb{R}^d} \mathcal{N}(\mathbf{y}_i|\mu, \mathbf{T}) \mathcal{N}(\mu|\mu_h^{(i-1)}, c_h^{(i-1)}\mathbf{T}) \mathcal{W}(\mathbf{T}|\delta_h^{(i-1)}, \mathbf{V}_h^{(i-1)}) d\mu d\mathbf{T}$$

$$= \int_{S_{++}^d} \mathcal{W}(\mathbf{T}|\delta_h^{(i-1)}, \mathbf{V}_h^{(i-1)}) \left\{ \int_{\mathbb{R}^d} \mathcal{N}(\mathbf{y}_i|\mu, \mathbf{T}) \mathcal{N}(\mu|\mu_h^{(i-1)}, c_h^{(i-1)}\mathbf{T}) d\mu \right\} d\mathbf{T}$$

Evaluating the inner integral within the brackets:

$$\int_{\mathbb{R}^d} \mathcal{N}(\mathbf{y}_i|\mu, \mathbf{T}) \mathcal{N}(\mu|\mu_h^{(i-1)}, c_h^{(i-1)}\mathbf{T}) d\mu$$

$$\propto \det(\mathbf{T})^{1/2} \det(c_h^{(i-1)}\mathbf{T})^{1/2}$$

$$\times \int_{\mathbb{R}^d} \exp\left(-\frac{1}{2}\left[c_h^{(i-1)}(\mu - \mu_h^{(i-1)})^T \mathbf{T}(\mu - \mu_h^{(i-1)}) + (\mathbf{y}_i - \mu)^T \mathbf{T}(\mathbf{y}_i - \mu)\right]\right) d\mu$$

$$= \det(\mathbf{T})^{1/2} \det(c_h^{(i-1)}\mathbf{T})^{1/2}$$

$$\times \exp\left(-\frac{1}{2}\frac{c_h^{(i-1)}}{1 + c_h^{(i-1)}}(\mathbf{y}_i - \mu_h^{(i-1)})^T \mathbf{T}(\mathbf{y}_i - \mu_h^{(i-1)})\right)$$

$$\times \int \exp\left(-\frac{1 + c_h^{(i-1)}}{2}(\mu - \mathbf{b})^T \mathbf{T}(\mu - \mathbf{b})\right) d\mu$$

$$\propto \frac{\det(\mathbf{T})^{1/2} \det(c_h^{(i-1)}\mathbf{T})^{1/2}}{\det((1 + c_h^{(i-1)})\mathbf{T})^{1/2}} \exp\left(-\frac{1}{2}\mathrm{tr}\left(\mathbf{T}\left\{\frac{c_h^{(i-1)}}{1 + c_h^{(i-1)}}(\mathbf{y}_i - \mu_h^{(i-1)})(\mathbf{y}_i - \mu_h^{(i-1)})^T\right\}\right)\right)$$

$$= \left(\frac{c_h^{(i-1)}}{1 + c_h^{(i-1)}}\right)^{d/2} \det(\mathbf{T})^{1/2} \exp\left(-\frac{1}{2}\mathrm{tr}\left(\mathbf{T}\left\{\frac{c_h^{(i-1)}}{1 + c_h^{(i-1)}}(\mathbf{y}_i - \mu_h^{(i-1)})(\mathbf{y}_i - \mu_h^{(i-1)})^T\right\}\right)\right)$$

Using this closed-form expression for the inner integral, we further obtain:

$$
L_{i,h}(\mathbf{y}_i) \propto \left( \frac{c_h^{(i-1)}}{1+c_h^{(i-1)}} \right)^{d/2} \int_{S_{++}^d} \frac{\det(\mathbf{V}_h^{(i-1)})^{-\delta_h^{(i-1)}}}{2^{d\delta_h^{(i-1)}}\Gamma_d(\delta_h^{(i-1)})} \det(\mathbf{T})^{(\delta_h^{(i-1)}+1/2)-\frac{d+1}{2}}
$$

$$
\times \exp\left( -\frac{1}{2}\mathrm{tr}\left( \mathbf{T}\left\{ (\mathbf{V}_h^{(i-1)})^{-1} + \frac{c_h^{(i-1)}}{1+c_h^{(i-1)}}(\mathbf{y}_i-\mu_h^{(i-1)})(\mathbf{y}_i-\mu_h^{(i-1)})^T \right\} \right) \right) d\mathbf{T}
\tag{7}
$$

$$
\propto \left( \frac{c_h^{(i-1)}}{1+c_h^{(i-1)}} \right)^{d/2} \frac{\Gamma_d(\delta_h^{(i-1)}+\frac{1}{2})}{\Gamma_d(\delta_h^{(i-1)})}
$$

$$
\times \frac{\det(\mathbf{V}_h^{(i-1)})^{-\delta_h^{(i-1)}}}{\det\left( \left\{ (\mathbf{V}_h^{(i-1)})^{-1} + \frac{c_h^{(i-1)}}{1+c_h^{(i-1)}}(\mathbf{y}_i-\mu_h^{(i-1)})(\mathbf{y}_i-\mu_h^{(i-1)})^T \right\}^{-1} \right)^{-(\delta_h^{(i-1)}+\frac{1}{2})}}
$$

$$
= \left( r_h^{(i-1)} \right)^{d/2} \frac{\Gamma_d(\delta_h^{(i-1)}+\frac{1}{2})}{\Gamma_d(\delta_h^{(i-1)})} \frac{\det((\mathbf{V}_h^{(i-1)})^{-1})^{-1/2}}{\det\left( \mathbf{I}_d + r_h^{(i-1)}(\mathbf{y}_i-\mu_h^{(i-1)})(\mathbf{y}_i-\mu_h^{(i-1)})^T\mathbf{V}_h^{(i-1)} \right)^{\delta_h^{(i-1)}+\frac{1}{2}}}
$$

$$
= \left( r_h^{(i-1)} \right)^{d/2} \frac{\Gamma_d(\delta_h^{(i-1)}+\frac{1}{2})}{\Gamma_d(\delta_h^{(i-1)})} \frac{\det(\mathbf{V}_h^{(i-1)})^{1/2}}{\left( 1 + r_h^{(i-1)}(\mathbf{y}_i-\mu_h^{(i-1)})^T\mathbf{V}_h^{(i-1)}(\mathbf{y}_i-\mu_h^{(i-1)}) \right)^{\delta_h^{(i-1)}+\frac{1}{2}}}
\tag{8}
$$

$$
= \left( \frac{r_h^{(i-1)}}{2\delta_h^{(i-1)}} \right)^{d/2} \frac{\Gamma_d(\delta_h^{(i-1)}+\frac{1}{2})}{\Gamma_d(\delta_h^{(i-1)})} \frac{\det((\mathbf{\Sigma}_h^{(i-1)})^{-1})^{1/2}}{\left( 1 + \frac{r_h^{(i-1)}}{2\delta_h^{(i-1)}}(\mathbf{y}_i-\mu_h^{(i-1)})^T(\mathbf{\Sigma}_h^{(i-1)})^{-1}(\mathbf{y}_i-\mu_h^{(i-1)}) \right)^{\delta_h^{(i-1)}+\frac{1}{2}}}
\tag{9}
$$

where we used the determinant identity $\det(\mathbf{I}+\mathbf{a}\mathbf{b}^T\mathbf{M}) = 1 + \mathbf{b}^T\mathbf{M}\mathbf{a}$ in the last step. We also defined $r_h^{(i)} := \frac{c_h^{(i)}}{1+c_h^{(i)}}$ and used $\mathbf{V}_h^{(i)} = \frac{(\mathbf{\Sigma}_h^{(i)})^{-1}}{2\delta_h^{(i)}}$.

## 3 Appendix C

**Theorem.** *The following asymptotic behavior holds:*

$$
\lim_{n\to\infty} \frac{\log \prod_{j=1}^{n-1}(1+\frac{\alpha}{j})}{\alpha \log n} = 1.
$$

*Proof.* It is sufficient to establish the limit for $\lim_{N\to\infty} \sum_{k=m}^{N} \log(1+\alpha/k)/\log N$ for fixed $m$. Choose $m$ such that $|\alpha| < m-1$ and use $\log(1-x) = \sum_{k=1}^{\infty} x^k/k$ for $|x| < 1$ to get

$$
\sum_{k=m}^{N} \log\left(1+\frac{\alpha}{k}\right) = \sum_{l=1}^{\infty}(-1)^{l+1}\frac{\alpha^l}{l}\sum_{k=m}^{N}\frac{1}{k^l}.
\tag{10}
$$

Separate (10) into two terms:

$$
\sum_{l=1}^{\infty}(-1)^{l+1}\frac{\alpha^l}{l}\sum_{k=m}^{N}\frac{1}{k^l} = \alpha\sum_{k=m}^{N}\frac{1}{k} + \sum_{l=2}^{\infty}(-1)^{l+1}\frac{\alpha^l}{l}\sum_{k=m}^{N}\frac{1}{k^l}.
\tag{11}
$$

The first term is expressed in terms of the Euler-Mascheroni constant $\gamma_e$ as

$$
\sum_{k=m}^{N}\frac{1}{k} = \log N - \gamma_e - \sum_{k=1}^{m-1}\frac{1}{k} + o(1).
$$

Thus, dividing by $\log N$ and taking the limit $N \to \infty$ we have a limiting value of unity. The second term of (11) is bounded. To see this, use, for $l > 1$,

$$\sum_{k=m}^{\infty} \frac{1}{k^l} \leq \int_{m-1}^{\infty} \frac{dx}{x^l} = \frac{1}{l-1}(m-1)^{-(l-1)}.$$

Then the second term of (11) is bounded by

$$\sum_{l=2}^{\infty} \frac{\alpha^l}{l} \sum_{k=m}^{\infty} \frac{1}{k^l} \leq \sum_{l=2}^{\infty} \frac{\alpha^l}{l(l-1)}(m-1)^{-(l-1)}$$

$$= (m-1) \sum_{l=2}^{\infty} \frac{1}{l(l-1)} \left( \frac{\alpha}{m-1} \right)^l < \infty.$$

The result follows since the second term, being bounded, vanished when dividing by $\log N$ and taking the limit $N \to \infty$. □

## 4 Appendix D

**Lemma.** *Let $r_n$ and $\bar{r}_n$ be random sequences with the update laws*

$$\begin{aligned} P(r_{n+1} = r_n + 1) &= \tau_n \\ P(r_{n+1} = r_n) &= 1 - \tau_n \end{aligned}$$

*and*

$$\begin{aligned} P(\bar{r}_{n+1} = \bar{r}_n + 1) &= \sigma_n \\ P(\bar{r}_{n+1} = \bar{r}_n) &= 1 - \sigma_n, \end{aligned}$$

*and assume $\sigma_n \geq \tau_n$ for all $n \geq 1$ and that $\bar{r}_0 = r_0 = 0$. Then $\mathbb{E}[\bar{r}_n] \geq \mathbb{E}[r_n]$ for all $n \geq 1$.*

*Proof.* We first use induction to show that $P(r_n > t) \leq P(\bar{r}_n > t)$ holds for all $n$.

The base case is trivial because $r_0 = \bar{r}_0$. We next prove that given

$$P(r_n > t) \leq P(\bar{r}_n > t) \tag{12}$$

for a particular $n$ and all $t \in \mathbb{N}$, the same inequality holds for $n + 1$. We have

$$\begin{aligned} P(r_{n+1} > t) &= (1 - \tau_n)P(r_n > t) + \tau_n P(r_n > t - 1) \\ &\leq (1 - \tau_n)P(\bar{r}_n > t) + \tau_n P(\bar{r}_n > t - 1) \\ &\leq (1 - \sigma_n)P(\bar{r}_n > t) + \sigma_n P(\bar{r}_n > t - 1) \\ &= P(\bar{r}_{n+1} > t), \end{aligned} \tag{13}$$

where we used the inductive hypothesis (12) and the inequality $P(\bar{r}_n > t) \leq P(\bar{r}_n > t - 1)$. Thus, by induction, the inequality (12) holds for all $n$. Using (12), we further obtain:

$$\mathbb{E}[r_n] = \int_0^{\infty} P(r_n > t)dt \leq \int_0^{\infty} P(\bar{r}_n > t)dt = \mathbb{E}[\bar{r}_n]$$

The proof is complete. □

## 5 Appendix E

**Theorem.** *Let $\tau_n$ be a sequence of real-valued random variables $0 \leq \tau_n \leq 1$ satisfying $\tau_n \leq \frac{r_n+1}{a_n}$ for $n \geq N$, where $a_n = l_{n+1}(\mathbf{y}_{n+1})^{-1}n(\lambda + \log n)$, and where the nonnegative, integer-valued random variables $r_n$ evolve according to:*

$$P(r_{n+1} = k | r_n) = \begin{cases} \tau_n, & k = r_n + 1 \\ 1 - \tau_n, & k = r_n \end{cases}. \tag{14}$$

*Assume the following for $n \geq N$:*

1. $l_n(\mathbf{y}_n) \le \zeta$    (a.s.)

2. $D(p_T \parallel \tilde{L}_{n,K+}) \le \delta$    (a.s.)

*where $D(p \parallel q)$ is the Kullback-Leibler divergence between distributions $p(\cdot)$ and $q(\cdot)$. Then, as $n \to \infty$,*

$$r_n = O_P(\log^{1+\zeta\sqrt{\delta/2}} n) \tag{15}$$

$$\alpha_n = O_P(\log^{\zeta\sqrt{\delta/2}} n) \tag{16}$$

*Proof.* We can study the generalized Polya urn model in the slightly modified form:

$$P(\bar{r}_{n+1} = k | \bar{r}_n) = \begin{cases} \frac{\bar{r}_n+1}{a_n}, & \text{if } k = \bar{r}_n + 1 \\ 1 - \frac{\bar{r}_n+1}{a_n}, & \text{if } k = \bar{r}_n \end{cases} \tag{17}$$

Taking the conditional expectation of $\bar{r}_{n+1}$ with respect to the filtration $\mathcal{F}_{n+1} \overset{\text{def}}{=} \sigma(\bar{r}_1, \ldots, \bar{r}_n, \gamma_1, \ldots, \gamma_{n+1}, \mathbf{y}_1, \ldots, \mathbf{y}_{n+1})$, we get $\mathbb{E}[\bar{r}_{n+1}|\mathcal{F}_{n+1}] = (\bar{r}_n + 1)\left(1 + \frac{1}{a_n}\right) - 1$. Set $x_n := \bar{r}_n + 1$. Rewriting this and using the definition of $a_n$, we obtain:

$$\mathbb{E}\left[x_{n+1}\Big|\mathcal{F}_{n+1}\right] \le x_n \left(1 + \frac{l_{n+1}(\mathbf{y}_{n+1})}{n \log n}\right) \tag{18}$$

Next, we seek an upper bound on the conditional expectation $\mathbb{E}[l_k(\mathbf{y}_k)|\mathcal{F}_{k-1}]$. This quantity can be bounded using convex duality Seeger (2003):

$$\mathbb{E}[l_k(\mathbf{y}_k)|\mathcal{F}_{k-1}] \le 1 + \frac{1}{s}D(p_T \parallel \tilde{L}_{k,K+}) + \frac{1}{s}\log \mathbb{E}_{\tilde{L}_{k,K+}}[e^{s(l_k(\mathbf{y}_k)-1)}]$$

For $k \ge N$, $l_k(\mathbf{y}_k) \le \zeta$ and $\mathbb{E}_{\tilde{L}_{k,K+}}[l_k(\mathbf{y}_k)] = 1$. By Hoeffding's inequality, $\mathbb{E}_{\tilde{L}_{k,K+}}[e^{s(l_k(\mathbf{y}_k)-1)}] \le e^{s^2\zeta^2/8}$. Using this bound, we obtain for $k \ge N$, $\mathbb{E}[l_k(\mathbf{y}_k)|\mathcal{F}_{k-1}] \le 1 + \delta/s + s\zeta^2/8$. Minimizing this as a function of $s > 0$, we obtain:

$$\mathbb{E}[l_k(\mathbf{y}_k)|\mathcal{F}_{k-1}] \le 1 + \zeta\sqrt{\frac{\delta}{2}} \tag{19}$$

Next, we upper bound $\mathbb{E}[x_{n+1}|\mathcal{F}_N]$ recursively. Taking the conditional expectation of both sides of (18), we obtain:

$$\mathbb{E}\left[x_{n+1}\Big|\mathcal{F}_n\right] \le \mathbb{E}\left[x_n\left(1 + \frac{l_{n+1}(\mathbf{y}_{n+1})}{n\log n}\right)\Big|\mathcal{F}_n\right] \tag{20}$$

We note that the function $l_{n+1}(\cdot)$ is $\mathcal{F}_n$-measurable. This follows since by definition, $l_{n+1}(\cdot) = \frac{L_0(\cdot)}{\sum_{h=1}^{k_n}\frac{m_n(h)}{n}L_{n+1,h}(\cdot)}$, and $m_n(h) = \sum_{l=1}^n I(\gamma_l = h)$ and $L_{n+1,h}(\cdot)$ are both $\mathcal{F}_n$-measurable (due to the parameter updates and (9)). Also note that $x_n = \bar{r}_n + 1$ is randomly determined by a biased coin flip given $\mathcal{F}_n$, increasing by 1 with probability $\frac{x_{n-1}}{a_{n-1}}$ and staying the same with probability $1 - \frac{x_{n-1}}{a_{n-1}}$. Since $a_{n-1}$ is $\mathcal{F}_n$-measurable, it follows that $x_n$ and $l_{n+1}(\mathbf{y}_{n+1})$ are conditionally independent given the history $\mathcal{F}_n$. Using this conditional independence, we obtain from (20):

$$\mathbb{E}\left[x_{n+1}\Big|\mathcal{F}_n\right] \le \mathbb{E}[x_n|\mathcal{F}_n]\left(1 + \frac{\mathbb{E}[l_{n+1}(\mathbf{y}_{n+1})|\mathcal{F}_n]}{n\log n}\right) \le \mathbb{E}[x_n|\mathcal{F}_n]\left(1 + \frac{1 + \zeta\sqrt{\delta/2}}{n\log n}\right) \tag{21}$$

where we used the bound (19) in the last inequality. Repeatedly conditioning and using (18) and (21): $\mathbb{E}[x_{n+1}|\mathcal{F}_N] \le \prod_{k=N}^n \left(1 + \frac{1+\zeta\sqrt{\frac{\delta}{2}}}{k\log k}\right)\mathbb{E}[x_N|\mathcal{F}_N] \le C_0 N \log^{1+\zeta\sqrt{\delta/2}} n$, where we used the Lemma in Appendix F and $C_0 = C(1 + \zeta\sqrt{\delta/2}, N)$, $x_N \le N$ in the last inequality. Taking the unconditional expectation and using $\mathbb{E}[r_n + 1] \le \mathbb{E}[\bar{r}_n + 1]$ (see Appendix D) yields the bound $\mathbb{E}[r_n + 1] \le C_0 N \log^{1+\zeta\sqrt{\delta/2}} n$. Markov's inequality then yields $\mathbb{P}\left(\frac{r_n+1}{C_0 N \log^{1+\zeta\sqrt{\delta/2}} n} > K\right) \le \frac{1}{K}$ which implies (15) by taking $K \to \infty$. Since $\alpha_n = \frac{r_n+1}{\lambda+\log n}$, the bound in (16) follows from a similar argument. The proof is complete.

$\square$

## 6  Appendix F

**Lemma.** *The following upper bound holds with constant $C(\phi, N) = e^{\frac{\phi}{N \log N}} / \log^\phi N$:*

$$\prod_{k=N}^{n} \left( 1 + \frac{\phi}{k \log k} \right) \le C(\phi, N) \log^\phi n$$

*Proof.* Using the elementary inequality $\log(1 + x) \le x$ for $x > -1$, we obtain:

$$\log \left( \prod_{k=N}^{n} \left( 1 + \frac{\phi}{k \log k} \right) \right) = \sum_{k=N}^{n} \log \left( 1 + \frac{\phi}{k \log k} \right)$$

$$\le \sum_{k=N}^{n} \frac{\phi}{k \log k} \le \phi \left( \int_N^n \frac{dx}{x \log x} + \frac{1}{N \log N} \right)$$

$$= \phi \left( \int_{\log N}^{\log n} \frac{dt}{t} + \frac{1}{N \log N} \right)$$

$$= \log \left( \frac{\log^\phi n}{\log^\phi N} \right) + \frac{\phi}{N \log N}$$

Taking the exponential of both sides yields the desired inequality. $\square$