[Reviews · NeurIPS 2015]

Submitted by Assigned_Reviewer_1

This paper introduces ASUGS (adaptive sequential updating and greedy search), building on the previous work on SUGS by Wang & Dunson 2011, which is a sequential (ie online) MAP inference method for DPMMs.

The main contribution of the paper is to provide online updating for the concentration parameter, \alpha.

The paper shows that the posterior distribution on \alpha can be expected to behave has a gamma distribution (that depends on the current number of clusters and on n) in the large-scale limit, assuming an exponential prior on \alpha.

ASUGS uses the mean of this gamma distribution as the \alpha for updating cluster assignments, the remainder of the algorithm proceeding as in SUGS (ie using conjugacy to update model parameters in an online fashion, with hard assignments of data to clusters.)

The paper also shows that this choice of \alpha is bounded by \log^\epsilon n for an arbitrarily small \epsilon, so that we may expect this process to converge, or at the very least be stable even in large settings.

Finally, the experimental results are convincing, showing an improvement over SVA (Lin 2013) on synthetic data.

The real data experiment demonstrates a qualitative improvement on MNIST digits.

This is a missed opportunity, I believe, it should be possible to construct a quantitative result here, by simply testing each methods predictive accuracy on held-out data?

I think this paper takes an important approach to inference in large scale Bayesian Nonparametric models, in that looking at the asymptotic behavior of hyperparameter posteriors may be more fruitful simply fixing them in advance, grid search, cross validation, or attempting to infer them with variational approximations.

It leads to simple computation that is suitable for the large scale setting with the potential of suffering less from approximation error.
Summary: An interesting and rigorously justified paper on large scale inference for DPMMs.

Submitted by Assigned_Reviewer_2

- 'Theorem 1', is a well-know result in the field. It should be easy to find references from any introductory lecture notes in the field. There seems to be nothing original about the proof, and proper attribution should be made anyways.

- SUGS updates parameters (step 2) but they are never used since the parameters are marginalized via conjugacy in step 1. This could be misleading.

- Moreover, the following argument starting at line 221 is hand-wavy and should be formalized since the following depends on it.

- It should be clearly pointed out that the mean of the posterior on alpha as a point estimates is an ad-hoc, non-Bayesian procedure.

- The argument in Section 4 assumes a parametric model. This should be clearly acknowledged as a limitation of the argument. Moreover the assumption 1 in the theorem seems fairly strong. The following paragraph mentions that they hold numerically, but the results do not seem very conclusive (given the large peak towards the end of the run). One or way or the other, the language should be changed, as it seems to suggest that their numerical results formally establish that the condition holds, which can never be strictly the case for an a.s. statement.

- Claims of 'stability' (in the title of section 4): what is established in this section is that the number of cluster will not grow too fast under strong assumptions. This is not the key issue: one could just put a hard limit of alpha K1 log(n) + K2 if really needed. What one would expect stability to mean is a bound on the closeness of the greedy posterior to the true posterior that do not blow up exponentially in the number of step size. It is unclear that ASUGS has this property even from the numerical example given the small datasets considered.

- There are large portions of the work that reproduce known results in the field. For example, section 5 is about standard results on matrix t-distributions, and much of the supplement contains standard, introductory level results in conjugacy. Unless there is something new there, a citation to existing in-depth tutorials (e.g. http://www.cs.ubc.ca/~murphyk/Papers/bayesGauss.pdf) would be more useful.

- The evaluation is clearly rushed and incomplete. First, since the motivation is computational, a key missing piece is some experiments that provides information on the trade-off between posterior approximation quality vs. compute cost.

- At least one classical, non-sequential methods should be included for completeness, e.g. the split-merge method of R. Neal.

- Since an important motivation of the paper is scalability, it would be good to use the full 60k in Section 6.2.

- There are not much useful information or insight gained in Section 6.2. More convincing data analysis would strengthen the paper.

- Code availability should be clarified. More info on the experimental setup should be added at the very minimum for reproducibility purpose.

Minor presentation issue: 'define the auxiliary variables': I would avoid calling these auxiliary variables, since auxiliary variables have a distinct, specialized meaning in computational Bayesian statistics.

Summary: The paper proposes an extension of a recently proposed method ('sequential updating and greedy search (SUGS)'), where the greediness of the algorithm is extended to the concentration parameter alpha. I have some concerns with respect to both the theoretical analysis and the experiments.

Submitted by Assigned_Reviewer_3

The paper proposes a sequential low-complexity inference procedure for Dirichlet process mixtures of Gaussians model for online clustering when the number of clusters are unknown apriori. The contribution of the paper is a minor modification to SUGS algorithm proposed in Wang and Dunson (2011) to adapt the concentration parameter \alpha with each step of sequential sampling.

It would be good to include results with SUGS-PM in Section 6.1 for fair comparison.

Minor comments: On line 40, Monte-Carlo Markov chain -> Markov chain Monte Carlo

Fix citation style to cite in parenthesis were necessary, e.g. line 53, 124, 159.

Incomplete sentence @ lines 337-339.

Summary: The contribution of the paper is a minor modification to SUGS algorithm proposed in Wang and Dunson (2011) to adapt the concentration parameter \alpha with each step of sequential sampling.

Author Feedback
Author rebuttal: We thank the reviewers for their constructive feedback, specifically for affirming the strong contributions of this paper.

Reviewer 1

We thank the reviewer for affirming that our paper is interesting and that the analysis presented provides a rigorous justification of the results. We remark that the approximation error in the asymptotic gamma distribution approximation of the posterior on alpha is fairly small in our experiments. We may add the predictive accuracy on held-out data for the MNIST data set if space permits in the final version.

Reviewer 2

Thm. 1 is certainly known, but is elevated to the level of a theorem since it motivates the updating procedure for the concentration parameter. A proof in the supplementary material is provided for completeness. A reference can also be cited if the reviewer provides one.

We believe that although the conjugacy calculations in the supplementary material can be considered standard by an expert in the field, the readers that might not be as familiar with such techniques might find this material useful and is also there for completeness. The asymptotic normality results in Section 5 are novel however to the best of our knowledge.

By 'stability' in Section 4, we mean that under our adaptive algorithm and model, the sequence constructed by the evolution of alpha's is asymptotically well behaved (Theorem 2). Stability in the sense of the paper can be enforced by cutting off \alpha by K1*log(n) + K2, but we have no basis for choosing this model or its parameters or even whether such a cut off is needed without some analysis of the type presented. The boundedness of l_n(y) is a strong assumption, but one that is validated in our simulations. The spikes in l_n(y) occur when data arrives from 'as yet unseen' classes. The y-axis scale will be increased in Fig. 2 in the final revision because the initial spikes are the most dominant ones, while the remaining ones are small when relatively compared to the first few dominant ones (which is when the learning occurs). It is worth noting that l_n(y) \leq 1/\epsilon for a modification of the original problem that replaces the likelihood f(y|\theta) with a mixture model of the form (1-\epsilon) f(y|\theta) + \epsilon L_0(y) (see footnote 1 in paper). Thus, problems close to the one posed satisfy the boundedness hypothesis.

Since it was shown in the SVA paper by Lin (2013) that SVA outperforms other non-sequential inference methods on synthetic and real data sets, we focus on improving upon SVA in our paper. Further experimental tradeoffs may be added in future work; we stress that the main contributions in this paper is building a theoretical analysis for adaptive sequential inference based on asymptotics.

MATLAB code will be made available online upon acceptance of this paper.

Reviewer 3

We disagree with the reviewer that this is a minor modification to the SUGS algorithm proposed in Wang & Dunson (2011). We provide a novel way to adaptively choose the concentration parameter based on a rigorous analysis. Furthermore, we prove growth rates on the number of classes that arise from such an adaptive algorithm, while Wang & Dunson (2011) do not provide any such analysis.

In addition, our analysis allows for an easily computable update of the concentration parameter, unlike the computationally expensive sequential Bayesian design of Wang & Dunson (2011) (see discussion in Section 2.1 of our paper). Experimental results are also provided to show the benefit of our methods.

SUGS-PM may be added in the final version of the paper for completeness. However, we note that it would resort to many prune-and-merge steps to clean up the model because the non-adaptive concentration parameter choice would lead to a creation of many superfluous classes (or might underestimate the true number of classes in which case prune-merge steps do not help).